# Comparative Analysis of Root Canal Dentin Removal Capacity of Two NiTi Endodontic Reciprocating Systems for the Root Canal Treatment of Primary Molar Teeth. An In Vitro Study

**DOI:** 10.3390/jcm11020338

**Published:** 2022-01-11

**Authors:** Vicente Faus-Llácer, Dalia Pulido Ouardi, Ignacio Faus-Matoses, Celia Ruiz-Sánchez, Álvaro Zubizarreta-Macho, Anabella María Reyes Ortiz, Vicente Faus-Matoses

**Affiliations:** 1Department of Stomatology, Faculty of Medicine and Dentistry, University of Valencia, 46010 Valencia, Spain; fausvj@uv.es (V.F.-L.); dapuou@alumni.uv.es (D.P.O.); ignacio.faus@uv.es (I.F.-M.); ceruizsan@gmail.com (C.R.-S.); vicente.faus@uv.es (V.F.-M.); 2Department of Endodontics, Faculty of Health Sciences, Alfonso X El Sabio University, 28691 Madrid, Spain; 3Department of Surgery, Faculty of Medicine and Dentistry, University of Salamanca, 37008 Salamanca, Spain; 4Department of Implant Surgery, Faculty of Health Sciences, Alfonso X El Sabio University, Avenida Universidad, 1, Villanueva de la Cañada, 28691 Madrid, Spain; 5Department of Pediatric Dentistry, Faculty of Health Sciences, Alfonso X El Sabio University, 28691 Madrid, Spain; areyeort@uax.es

**Keywords:** endodontics, endodontic reciprocating file, micro-computed tomography scan, root canal dentin removal, primary molar teeth

## Abstract

The objective of the present study was to evaluate and compare the dentin removal capacity of Endogal Kids and Reciproc Blue NiTi alloy endodontic reciprocating systems for root canal treatments in primary second molar teeth via a micro-computed tomography (micro-CT) scan. Materials and Methods: Sixty root canal systems in fifteen primary second molar teeth were chosen and classified into one of the following study groups: A: EK3 Endogal Kids (*n* = 30) (EDG) and B. R25 Reciproc Blue (*n* = 30) (RB). Preoperative and postoperative micro-CT scans were uploaded into image processing software to analyze the changes in the volume of root canal dentin using a mathematical algorithm that enabled progressive differentiation between neighboring pixels after defining and segmenting the root canal systems in both micro-CT scans. Volumetric variations in the root canal system and the root canal third were calculated using a *t*-test for independent samples or a nonparametric Mann–Whitney–Wilcoxon test. Results: Statistically significant differences (*p* = 0.0066) in dentin removal capacity were found between the EDG (2.89 ± 1.26 mm^3^) and RB (1.22 ± 0.58 mm^3^) study groups for the coronal root canal third; however, no statistically significant differences were found for the middle (*p* = 0.4864) and apical (*p* = 0.6276) root canal thirds. Conclusions: Endogal and Reciproc Blue NiTi endodontic reciprocating systems showed similar capacity for the removal of root canal dentin, except for the coronal root canal third, in which the Reciproc Blue NiTi endodontic reciprocating system preserved more root canal dentin tissue.

## 1. Introduction

The presence of bacteria within the root canal system poses a risk factor for the appearance of pulp and periapical diseases in both primary and permanent dentition [1,2]. The biomechanical preparation of the root canal system is therefore considered to be a fundamental step in the root canal treatment process in order to adequately eliminate bacteria, necrotic tissue, and infected dentin [3]. In addition, the root canal system must be funnel shaped, becoming narrower in the apical direction in order to maintain the original anatomy and enable sufficient obturation [4,5]. A pulpectomy is widely recommended for primary teeth so as to preserve arch length, maintain primary teeth, including their functional and aesthetic properties, and guide the proper eruption of permanent dentition [6]. Hand files are widely used to work with the root canal system in primary dentition; however, the root anatomy makes successful endodontic treatment difficult [7,8]. In primary teeth, the root canal system is characterized by high anatomical variability, including accessory and curved canals, as well as physiological root resorption that can alter the formation of the root canal system [9]. Nickel–titanium (NiTi) endodontic rotary instruments enable clinicians to maintain the original anatomy of curved canals, reducing the likelihood of potential mishaps during root canal system preparation [10]. Recently, novel NiTi endodontic rotary files have been specifically developed for the root canal treatment of primary teeth. Endogal Kids Rotary can be used either with a rotary or reciprocating motion; however, this latter movement is recommended for use in children, since it reduces the working time. This endodontic reciprocating system is manufactured in a NiTi alloy with heat treatment, and has a 17 mm length, 4% taper, 300µm apical diameter, and triangular cross-section design. Moreover, Reciproc Blue NiTi endodontic pediatric files also performs a reciprocating motion and is manufactured in a CM-Blue Wire NiTi alloy with heat treatment, and has a 17 mm length, 300µm apical diameter, and double-S cross-section design. The heat treatment improves the physical properties of NiTi endodontic rotary files, increasing their cyclic fatigue resistance and helping them adapt to different curvatures and angulations. Some studies have described the use of single files in a reciprocating motion for the root canal treatment of primary molars and reported significant advantages in pediatric dentistry, such as a decrease in working time, low risk of iatrogenic errors, or the prevention of cross-contamination [11,12,13]. That being said, root canal treatments can be affected by various factors, including anatomical design, diameter, kinematics, taper, and the number of files used during the procedure [14,15]. In addition, several techniques have been used to measure the amount of dentin removal, including plastic models, histologic sections, serial sectioning, scanning electron microscopic studies, radiographic comparison, and the silicone impression of un-instrumented root canal systems [15]. However, few studies have used the micro-CT with primary molars, which is a conservative, accurate, and nondestructive measurement procedure [11]. Micro-computed tomography (micro-CT) analysis has become a conservative measurement technique for obtaining an accurate 3D analysis, enabling both the quantitative and qualitative assessment of the root canal system anatomy after the shaping procedures [6,16].

The objective of the present study was to evaluate and compare the dentin removal capacity of Endogal Kids and Reciproc Blue NiTi endodontic reciprocating systems for the root canal treatment of primary second molar teeth via a micro-CT scan, with a null hypothesis (H_0_) that there are no differences in root dentin removal capacity between the Endogal Kids and Reciproc Blue NiTi endodontic reciprocating systems for root canal treatments in primary molar teeth.

## 2. Materials and Methods

### 2.1. Study Design

Sixty root canal systems were chosen from a total of fifteen primary second molar teeth (8 upper and 7 lower) that had been extracted for orthodontic or restorative reasons. Between January and March 2021, around three root canal systems were selected for study from cases at the Department of Stomatology at the University of Valencia in Valencia, Spain. All of the selected root canal systems presented no prior root canal filling materials or root resorption. A power of 80.00% was calculated using the bilateral Student’s *t*-test for two independent samples. When used to calculate the variation from the null hypothesis H₀: μ₁ = μ₂, the significance level of 5.00% and power of 80.00% meant that 60 root canal systems were necessary for the purposes of this study. The study was carried out as a randomized controlled experimental trial, in keeping with the norms outlined by the statement of the German Ethics Committee on the use of organic tissues as part of medical research (Zentrale Ethikkommission, 2003). Additionally, the study was reviewed and approved by the Ethics Committee of the University of Valencia under Process No. H1512122849636. All study participants provided their prior informed consent for participation in this study.

### 2.2. Experimental Procedure

The sixty root canal systems in the fifteen selected primary second molar teeth were assigned randomly (Epidat 4.1, Galicia, Spain) to one of the following NiTi endodontic reciprocating systems: A. EK3 Endogal Kids (Endogal, Galician Endodontics Company, Lugo, Spain) (*n* = 30) (EDG) or B. R25 Reciproc Blue (VDW, Baillagues, Switzerland) (*n* = 30) (RB). Impressions of the teeth were taken using polyvinyl siloxane material (Ref.: 7000054992, Express™ 2 Putty Soft, 3M ESPE™, Saint Paul, MN, USA) to enable the access cavity to be prepared using the technique described by Rover et al. [17]. The root canal working length was determined with a stainless steel #10 K-file (Dentsply Maillefer, Ballaigues, Switzerland) and observed under magnification (OPMI pico, Zeiss Dental Microscopes, Oberkochen, Germany) until the far end of the file became visible through the epical foramen. Each root canal system was manually prepared with up to a #25 K-file (Dentsply Maillefer, Ballaigues, Switzerland) before being performed upon according to the NiTi endodontic reciprocating system to which it had been assigned. Root canal systems randomly assigned to the EDG study group were prepared with a reciprocating movement, and the root canal systems randomly assigned to the RB study group were also prepared with a reciprocating motion. In addition, the root canal systems were irrigated using a 5 mL sterile saline solution (Braun, Jaén, Spain) with 5 mL of 17% EDTA (SmearClear; SybronEndo, CA, USA) and 5 mL of 5.25% NaOCl (Clorox; Oakland, CA, USA), administered using a 0.3 mm endodontic needle (Miraject Endo Luer; Hager & Werken, Duisburg, Germany) inserted into the working length up to 1 mm. The teeth were kept in an incubator (mco-18aic, Sanyo, Moriguchi, Osaka, Japan) and stored at 37 °C with 100% relative humidity. A single clinician performed all the root canal procedures.

### 2.3. Micro-CT Scanning

Preoperative and postoperative micro-CT scans (Micro-CAT II, Siemens Preclinical Solutions, Knoxville, TN, USA) were performed to analyze and compare the amount of root canal dentin removed by the Endogal Kids and Reciproc Blue NiTi endodontic reciprocating systems subsequent to the root canal treatment of the primary second molar teeth. The scans were taken using the following exposure parameters: 88 µA, 90 kV, 360° rotation, and 50 μm isotropic resolution. Tomographic 3D images of the entire tooth showed a total of 512 slices, with an isotropic voxel size of 50 microns and a 512 × 512-pixel resolution for each slice (Figure 1A–F).

### 2.4. Measurement Procedure

The analysis of the change in the volume of dentin removed after the root canal procedures was carried out using image processing software (ImageJ, National Institutes of Health, Bethesda, MD, USA) after the root canal systems had been defined and segmented (ROI: 10 × 10 × 10 mm) using the preoperative and postoperative micro-CT scans (Micro-CAT II, Siemens Preclinical Solutions, Knoxville, TN, USA). In addition, transverse section images were also analyzed in the apical, middle, and coronal root thirds (Figure 2).

### 2.5. Statistical Tests

Statistical analysis was carried out using SAS 9.4 (SAS Institute Inc., Cary, NC, USA). The mean and standard deviation (SD) were used for the descriptive analysis of quantitative data. For each of the variables, the difference between the pre- and postoperative values was analyzed using a t-test for independent samples or a nonparametric Mann–Whitney–Wilcoxon test based on compliance with the application criteria. *p* < 0.05 was determined to be the level for statistical significance.

## 3. Results

Table 1 and Figure 2 show the mean and standard deviation values for the volume of root canal system (mm^3^) between EDG and RB NiTi endodontic files at coronal, middle and apical root canal third.

The paired *t*-test found no statistically significant differences (*p* = 0.0767) in the volume of root canal dentin removed between the EDG (4.30 ± 2.58 mm^3^) and RB (2.32 ± 1.07 mm^3^) study groups. However, the paired *t*-test found statistically significant differences (*p* = 0.0066) between the EDG (2.89 ± 1.26 mm^3^) and RB (1.22 ± 0.58 mm^3^) study groups in the volume of root canal dentin removed at the coronal root canal third (Figure 3).

However, the paired *t*-test did not find any statistically significant differences (*p* = 0.4864) in the volume of root canal dentin removed between the EDG (1.20 ± 1.27 mm^3^) and RB (0.85 ± 0.47 mm^3^) study groups at the middle root canal third (Figure 4).

Moreover, the paired *t*-test did not reveal any statistically significant differences (*p* = 0.6276) in the volume of root canal dentin removed between the EDG (0.20 ± 0.25 mm^3^) and RB (0.26 ± 0.17 mm^3^) study groups at the apical root canal third (Figure 5).

## 4. Discussion

The results of the present study refute the null hypothesis (H_0_) that there is no difference in root dentin removal capacity between the Endogal Kids and Reciproc Blue NiTi endodontic reciprocating systems for the root canal treatment of primary molar teeth.

Various methods have previously been used to evaluate root canal instrumentation, including plastic models, serial sectioning, scanning electron microscopic studies, and radiographic comparisons [18]. More recently, noninvasive 3D techniques, such as CBCT or micro-CT scans, have been used to assess the efficiency of cleaning and dentin removal after root canal treatment procedures [19]. In addition, high-resolution 3D micro-CT images are the gold standard for evaluating the root canal system anatomy and root canal instrumentation [20,21]. In the present study, micro-CT scans were used to examine the internal anatomy of the root canal system and evaluate the effectiveness of root canal instrumentation on the root canal system of primary second molar teeth. The authors selected the primary second molars because the anatomy of this tooth is very similar to that of the permanent first molar, which allows a comparison to be made between them. In addition, the eruption chronology of the second premolars is usually later than that of the first premolars, which leads to less root resorption of the primary second molars compared to the primary first molars [22].

Micro-CT scan measurement techniques have previously been used to analyze the amount of root canal dentin removed from permanent teeth after root canal treatment. Yilmaz et al. reported no statistically significant differences between the amount of dentin removed by ProTaper Next (Dentsply Maillefer, Ballaigues, Switzerland), OneShape (MicroMega, Besançon, France), and EdgeFile (Edge Endo, Albuquerque, NM) NiTi alloy endodontic rotary files for the whole canal length (*p* > 0.05) [23]. Moreover, de Albuquerque et al. reported that the Protaper Next, Wave One Gold, Predesign Logic, and Vortex Blue NiTi alloy endodontic systems caused a greater dentin removal at the coronal third (9 mm), decreasing at the apical one (3 mm) [24]. These findings are aligned with the results shown in the present study.

The root canals in primary teeth are not always easy to shape and obturate during treatments. In fact, many characteristics of the root canal anatomy make endodontic treatment difficult, potentially resulting in apical transportation, zipping, perforations, or gaps [18,21]. Esentürk et al. observed that 60% of the root canal system was left un-instrumented upon after root canal preparation due to the anatomical complexity of the primary molars, highlighting a need for NiTi alloy endodontic rotary instruments to be developed for use in primary teeth [25]. Prabhakar et al. found that the Wave One NiTi alloy endodontic reciprocal system enabled quicker and safer instrumentation compared with the One Shape NiTi alloy endodontic rotary system, because the former reduces levels of both torsional and flexural stress, as well as the number of instruments required for the sequence [11]. According to their findings, Katge et al. reported that the Wave One NiTi alloy endodontic reciprocal system had a statistically greater cleaning capability than the Protaper NiTi alloy endodontic rotary system at the coronal and middle third due to the benefits of reciprocating motion [12]. However, the risk of root perforation and root canal transportation is more correlated with a high taper value than a reciprocating or continuous motion, which means that the NiTi alloy endodontic system should be selected primarily based on the taper [21]. Ramazani et al. assessed the efficiency of Mtwo NiTi alloy endodontic rotary files and Reciproc NiTi alloy endodontic reciprocating files when cleaning, finding no statistically significant differences between the two study groups, although the Reciproc NiTi alloy endodontic reciprocating files required less preparation time [13]. Azar et al. found no statistically significant differences in cleaning capabilities between Mtwo NiTi alloy endodontic rotary files, Protaper NiTi alloy endodontic rotary files, and manual K files in the three root thirds of the root canal system, measuring the differences using ink and stereo microscopes [26]. These results were corroborated by the findings of Ramazani et al. for Mtwo NiTi alloy endodontic rotary files and K files [13]; Moghaddam et al. for Master NiTi alloy endodontic rotary files, Rotary Flex NiTi alloy endodontic rotary files, and K files [8]; and Mehalawat et al. for Profile NiTi alloy endodontic rotary files and K files [27]. However, Madan et al. did observe statistically significant differences between Profile NiTi alloy endodontic rotary files and K files when using the same ink removal method, during which the Profile NiTi alloy endodontic rotary files were more efficient at cleaning the coronal root third, while the manual files were better at cleaning the apical root third [28].

Some studies have compared the cleaning capacity of both manual and NiTi alloy endodontic rotary files in permanent teeth [23], but not as many included primary teeth, and only a few of the studies used micro-CT scan assessments. The volume of dentin removed reveals the remaining dentin thickness, which is needed to provide enough resistance for root canal treatments. The force with which root canal instruments are used is in direct proportion to the amount of dentin removed [29]. Although manual instrumentation is commonly used in primary teeth, many studies have found that more dentin is removed using manual files than rotary instrumentation [19,20,29,30]. Selvakumar et al. used K3 NiTi alloy endodontic rotary files (with a 0.02 taper) and found significantly lower dentin removal when compared with manual K files and K3 NiTi alloy endodontic rotary files (with a 0.04 taper), which were shown to remove more dentin tissue in the coronal and apical root thirds in comparison with K files and K3 NiTi alloy endodontic rotary files (with a 0.02 taper) [19]. On the other hand, Zameer et al. observed no statically significant differences when using either the 2% or 4% taper rotary files to remove dentin, without damaging the dentinal walls and achieving an improved canal shape for root canal filling material [31]. In addition, Moghaddam found that a continuous rotation movement with up to a #30 apical diameter enabled better instrumentation and safer results when used with primary teeth without excessive dentin removal [8]. However, Zameer et al. observed a greater number of root perforations when dentin removal was performed using 4% taper NiTi alloy endodontic rotary files compared with 2% NiTi alloy endodontic rotary files and manual K files [31]. This result corroborates the findings of Kummer et al., who used rotary 6% taper NiTi alloy endodontic rotary files with a #30 apical diameter and found three root perforations, concluding that the mesial and distal roots of lower molars and mesiobuccal roots of upper molars had a higher risk of root perforation [30]. In addition, Barasuol et al. observed two perforations in the apical and middle root third, as well as root canal transportation, when using 8% taper Reciproc NiTi alloy endodontic reciprocating files [21]. Files with a larger taper can result in the reduced thickness of the dentinal wall, leading to greater fragility of the teeth and a higher risk of root perforation [20]. Madan et al. found that instrumentation failure was reduced when using 0.04 taper Profile NiTi alloy endodontic rotary files, which were also less damaging for primary teeth [28].

The strengths and innovation of the current study are that not many studies analyze the effect of specific pediatric instrumentation systems on primary teeth, even though pulpectomy is a widely performed dental treatment. Furthermore, the instrumentation systems compared are very novel; especially the Reciproc Blue system, which has not been released on the market. Finally, the micro-CT scan measurement technique for dentin removal analysis is very accurate and innovative.

The present findings are limited by the constraints of an in vitro study. The use of instrumentation with primary teeth is not subject to any universal guidelines, and clinical trials are needed to obtain clinical results. Additional studies should be carried out on a larger sample size, as well as using pediatric files.

## 5. Conclusions

To summarize, within the constraints of this in vitro study, the results indicate that the Endogal and Reciproc Blue NiTi endodontic reciprocating systems are similarly capable of removing root canal dentin, except for in the coronal root canal third, in which the Reciproc Blue NiTi endodontic reciprocating system preserved more root canal dentin tissue.

## Figures and Tables

**Figure 1 jcm-11-00338-f001:**
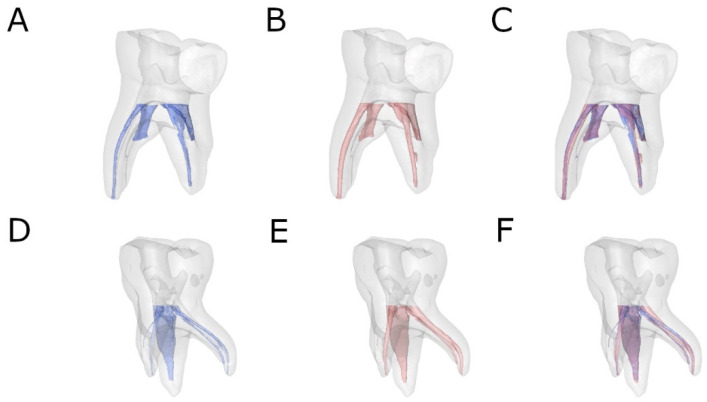
Reconstructed 3D micro-CT images of (**A**) preoperative (blue), (**B**) postoperative (red), and (**C**) superimposed pre- and postoperative images of the EDG study group (blue and red) and (**D**) preoperative (blue), (**E**) postoperative (red), and (**F**) superimposed pre- and postoperative images of the RB study group (blue and red).

**Figure 2 jcm-11-00338-f002:**
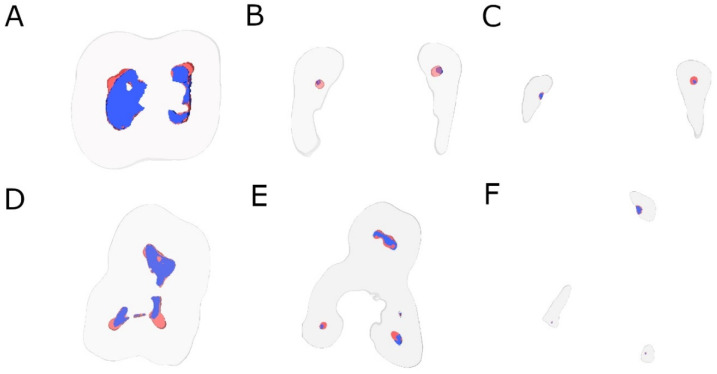
Transverse section images after aligning preoperative (blue) and postoperative (red) micro-CT scans of the EDG study group at the (**A**) coronal, (**B**) middle, and (**C**) apical root third and (**D**) coronal, (**E**) middle, and (**F**) apical root third of the RB study group.

**Figure 3 jcm-11-00338-f003:**
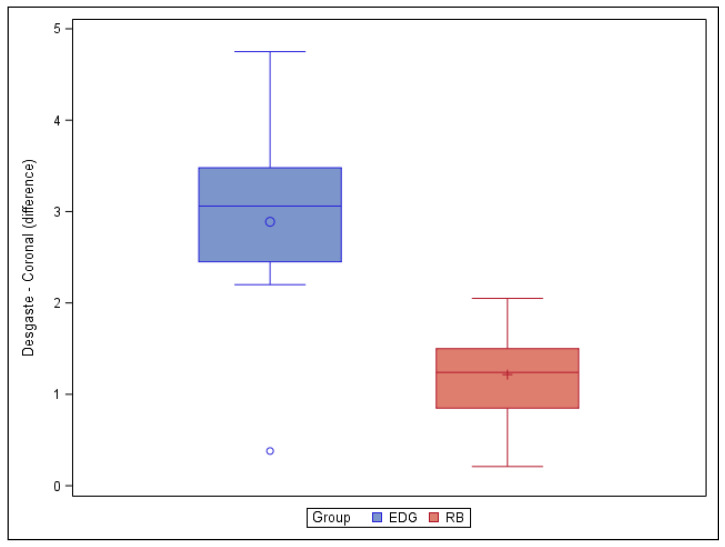
Box plot of the difference in dentin volume pre- and post-root canal procedure between the EDG and RB study groups at the coronal level.

**Figure 4 jcm-11-00338-f004:**
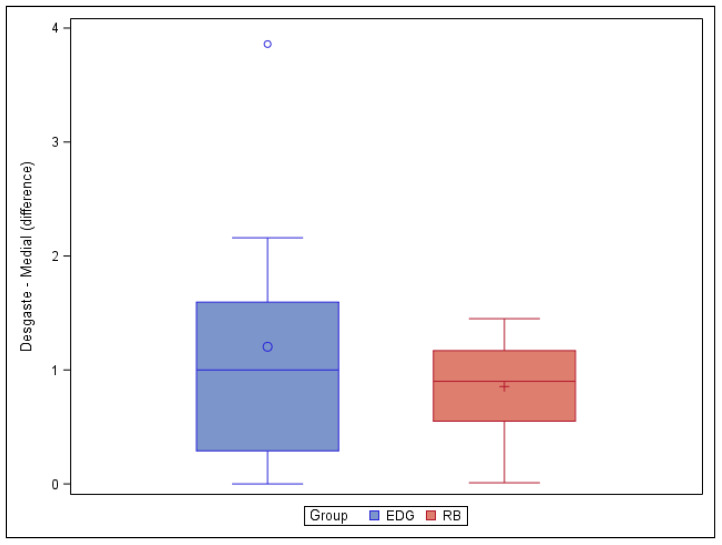
Box plot of the difference in dentin volume pre- and post-root canal procedure between the EDG and RB study groups at the middle level.

**Figure 5 jcm-11-00338-f005:**
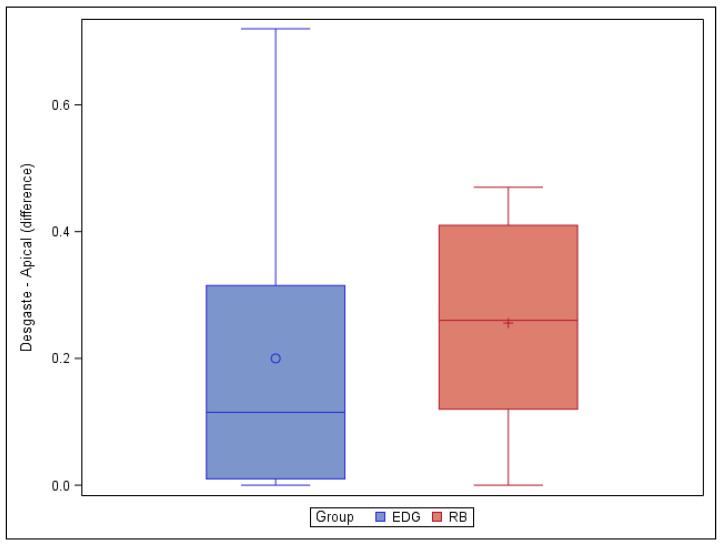
Box plot of the difference in dentin volume pre- and post-root canal procedure between the EDG and RB study groups at the apical level.

**Table 1 jcm-11-00338-t001:** Descriptive analysis of the volume of root canal system (mm^3^) between EDG and RB NiTi endodontic files at coronal, middle and apical root canal third.

Study Group	Root Third	Time	*n*	Mean	SD	Minimum	Maximum
EDG	Coronal	Preoperative	30	7.61 ^a^	4.81	3.58	18.20
		Postoperative	30	10.50 ^a^	5.78	4.13	22.95
	Middle	Preoperative	30	1.74 ^a^	1.23	0.22	4.20
		Postoperative	30	2.94 ^a^	1.58	0.83	5.81
	Apical	Preoperative	30	0.33 ^a^	0.36	0.00	1.17
		Postoperative	30	0.53 ^a^	0.39	0.00	1.19
RB	Coronal	Preoperative	30	6.53 ^b^	1.08	5.22	8.58
		Postoperative	30	7.75 ^b^	1.48	6.07	10.63
	Middle	Preoperative	30	1.71 ^a^	1.10	0.05	3.04
		Postoperative	30	2.56 ^a^	1.38	0.95	3.91
	Apical	Preoperative	30	0.40 ^a^	0.23	0.03	0.68
		Postoperative	30	0.66 ^a^	0.25	0.41	1.15

EDG: Endogal; RB: Reciproc Blue; ^a,b^: statistical significance.

## Data Availability

Information is available on request in accordance with any relevant restrictions (e.g., privacy or ethical).

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
