# Peer review of "Comparative Analysis of Root Canal Dentin Removal Capacity of Two NiTi Endodontic Reciprocating Systems for the Root Canal Treatment of Primary Molar Teeth. An In Vitro Study"

_jcm, 2022, doi:10.3390/jcm11020338_

Round 1

Reviewer 1 Report

comments are given in the below files 

Author Response

Dear Reviewer 1:

I’m pleased to resubmit the manuscript of the work entitled, “Comparative Analysis of Root Canal Dentin Removal Capacity of Two NiTi Endodontic Reciprocating Systems for the Root Canal Treatment of Primary Molar Teeth. An In Vitro Study”

Reviewer 1: Moderate English changes required

Response: In order to adapt to the reviewer's 1 comments, we have sent the manuscript to the English Editing Service of MDPI. We attached the Certificate.

Reviewer 1: kindly change the term to middle root canal third.

Response: In order to adapt to the reviewer's 1 comments, we have changed the word in the manuscript.

Reviewer 1: use appropriate term i.e primary teeth.

Response: In order to adapt to the reviewer's 1 comments, we have changed the word in the manuscript.

Reviewer 1: primary dentition.

Response: In order to adapt to the reviewer's 1 comments, we have we have changed the word in the manuscript.

Reviewer 1: spell check.

Response: In order to adapt to the reviewer's 1 comments, we have we have checked the word.

Reviewer 1: why is the teeth removed for restorative reason use along with orthodontic extraction. Since this can be a confounding factor. So kindly provide explanation.

Response: In order to adapt to the reviewer's 1 comments, we clarify that the teeth used in the study were extracted for orthodontic reasons and also for restorative reasons, because they presented cavities so extensive and deep that they could not be restored.

Reviewer 1: how can one assess the resorption rate in primary tooth indicated for rct or orthodontic extraction.

Response: In order to adapt to the reviewer's 1 comments, we clarify that the degree of root resorption of primary teeth was determined by a periapical radiograph performed to diagnose the tooth. However, restorative criteria were also taken into account in the diagnosis.

We take this opportunity to thank the recommendations and suggestions made by the reviewers to improve the document.

Yours sincerely,

Reviewer 2 Report

The submitted manuscript intend to evaluate the root canal dentin removal capacity of two NiTi endodontic reciprocating systems for the root canal treatment of primary molar teeth. It is a good experimental designed manuscript. The following points are suggested for futher consideration.

  1. It needs to state the rationales to choose primary second molar teeth in the current study.
  2. The innovations of the current study need to be stated more clearly.
  3. Does similar study has already been done for the adult teeth? It is worthwhile to mention this issue within the discussion section.

Author Response

Dear Reviewer 2,

I’m pleased to resubmit the manuscript of the work entitled, “Comparative Analysis of Root Canal Dentin Removal Capacity of Two NiTi Endodontic Reciprocating Systems for the Root Canal Treatment of Primary Molar Teeth. An In Vitro Study”

Reviewer 2: I don't feel qualified to judge about the English language and style

Response: In order to adapt to the reviewer's 2 comments, we have sent the manuscript to the English Editing Service of MDPI. We attached the Certificate.

Reviewer 2: It needs to state the rationales to choose primary second molar teeth in the current study

Response: In order to adapt to the reviewer's 2 comments, the authors selected the primary second molars because the anatomy of this tooth is very similar to that of the permanent first molar, which allows comparison between them. In addition, the eruption chronology of the second premolars is usually later than that of the first premolars, which leads to less root resorption of the primary second molars compared to the primary first molars. Finally, we select the primary second molars following the methodology of the following article: Azar MR, Mokhtare M. Rotary Mtwo system versus manual K-file instruments: efficacy in preparing primary and permanent molar root canals. Indian J Dent Res. 2011 Mar-Apr;22(2):363. doi: 10.4103/0970-9290.84283. We have explained the reason in the Discussion section.

Reviewer 2: The innovations of the current study need to be stated more clearly.

Response: In order to adapt to the reviewer's 2 comments, we clarify that the strengths and innovation of the current study are that there are not many studies that analyze the effect of specific pediatric instrumentation systems on primary teeth, even though pulpectomy is a widely performed dental treatment. Furthermore, the instrumentation systems compared are very novel; especially the Reciproc Blue system, which has not been released on the market. Finally, the micro-CT scan measurement technique for dentin removal analysis is very accurate and innovative. We have explained the reason in the Discussion section.

Reviewer 2: Does similar study has already been done for the adult teeth? It is worthwhile to mention this issue within the discussion section

Response: In order to adapt to the reviewer's 2 comments, we have clarified that micro-CT scan measurement technique has been previously used to analyze the amount of root canal dentin removed of permanent teeth after root canal treatment. Yilmaz et al reported no statistically significant differences between the amount of dentin removed of ProTaper Next (Dentsply Maillefer, Ballaigues, Switzerland), OneShape (MicroMega, Besançon, France), and EdgeFile (Edge Endo, Albuquerque, NM) NiTi alloy endodontic rotary files for the whole canal length (p > 0.05). Moreover, de Albuquerque et al reported that Protaper Next, Wave One Gold, Predesign Logic and Vortex Blue NiTi alloy endodontic systems caused a greater dentin removal at the coronal third (9 mm), decreasing at the apical one (3 mm). This findings are aligned with the results showed in the present study. We have explained the reason in the Discussion section.

We take this opportunity to thank the recommendations and suggestions made by the reviewers to improve the document.

Yours sincerely,